# Influence of Processing Glass-Fiber Filled Plastics on Different Twin-Screw Extruders and Varying Screw Designs on Fiber Length and Particle Distribution

**DOI:** 10.3390/polym14153113

**Published:** 2022-07-30

**Authors:** Annette Rüppel, Susanne Wolff, Jan Philipp Oldemeier, Volker Schöppner, Hans-Peter Heim

**Affiliations:** 1Institut für Werkstofftechnik, Kunststofftechnik, University of Kassel, 34125 Kassel, Germany; annette.rueppel@uni-kassel.de (A.R.); susanne.wolff@uni-kassel.de (S.W.); heim@uni-kassel.de (H.-P.H.); 2Kunststofftechnik Paderborn, Paderborn University, 33098 Paderborn, Germany; volker.schoeppner@ktp.upb.de

**Keywords:** extrusion, polypropylene, glass fiber, fiber reinforced, simulation, fiber shortening, compound, SIGMA, dynamic image analysis

## Abstract

Due to their valuable properties (low weight, and good thermal and mechanical properties), glass fiber reinforced thermoplastics are becoming increasingly important. Fiber-reinforced thermoplastics are mainly manufactured by injection molding and extrusion, whereby the extrusion compounding process is primarily used to produce fiber-filled granulates. Reproducible production of high-quality components requires a granulate in which the fiber length is even and high. However, the extrusion process leads to the fact that fiber breakages can occur during processing. To enable a significant quality enhancement, experimentally validated modeling is required. In this study, short glass fiber reinforced thermoplastics (polypropylene) were produced on two different twin-screw extruders. Therefore, the machine-specific process behavior is of major interest regarding its influence. First, the fiber length change after processing was determined by experimental investigations and then simulated with the SIGMA simulation software. By comparing the simulation and experimental tests, important insights could be gained and the effects on fiber lengths could be determined in advance. The resulting fiber lengths and distributions were different, not only for different screw configurations (SC), but also for the same screw configurations on different twin-screw extruders. This may have been due to manufacturer-specific tolerances.

## 1. Introduction

Due to their valuable properties, fiber-reinforced thermoplastics have a wide range of applications. Possible areas of application can be found in the fields of the automotive industry, aeronautic industry, and the household sector. In particular, polypropylene composites have been established as an alternative for a wide range of applications, due to their positive properties. In addition to their lightweight potential, the materials offer good mechanical and thermal properties, as well as a low weight and good mechanical strength, at low production costs [1,2,3,4]. As reinforcing materials, glass fibers are mainly used. This can significantly increase the mechanical properties (notched impact strength, tensile strength). Furthermore, the addition of glass fibers enhances the heat deflection temperature. In addition, the use of fibers is becoming increasingly popular, as the materials produced with them have a low density. Moreover, environmental awareness is becoming increasingly important, which is why the automotive industry is using fiber-reinforced materials in automotive interiors [5,6,7,8].

Fiber-reinforced thermoplastics are mainly obtained by injection molding and extrusion processes. However, an extrusion compounding process is primarily used to produce fiber-filled granulates, while the injection molding process is used to manufacture the products. In addition to the fiber orientation, process-related damage to the fibers (shortening of the fibers) can also have a decisive influence on the subsequent component properties. In particular, this can lead to undesirable changes in mechanical properties. [9,10,11,12].

In order to produce components with the best possible properties, a flawless granulate from the extrusion process is required for the subsequent injection molding process. One aspect to consider, as described previously, is that the original fiber length can be significantly reduced in some process steps during extrusion compounding. This can lead to a reduction in fiber reinforcement. Due to different extrusion lines and screw configurations, the process parameters may vary, leading to different resulting fiber lengths and compound properties. As a result, the subsequent component properties can be very different from each other.

The influence of extrusion parameters on fiber length reduction in a twin-screw extruder for fiber-reinforced plastics was previously fundamentally investigated. The change from a laboratory extruder to an industrial extruder led to differences in the fiber length distributions. With the industrial extruder, longer fibers and a more widely distributed fiber length could be identified in the compound. Under the conditions tested, the laboratory extruder was more fiber-damaging than the industrial extruder [13]. Gamon et al. investigated the influence of different parameters (rotational speed, feed rate, shear rate) in the extrusion process of natural fiber-reinforced PLA, concerning the fiber morphology (fiber breaks, fiber length change). Here, an increase in screw speed with a simultaneous increase in feed rate could contribute to maintaining the fiber length. In addition, it was shown that fiber breaks occur more frequently with longer fibers than with short fibers [14]. Durmaz et al. examined the properties of carbon fiber-reinforced bio-based polyamide in the extrusion and injection molding process. The fiber content ranged from 20 to 40 %. It was found that the average fiber length decreased significantly with increasing fiber content [15]. In their study, Wang et al. investigated fiber orientation, as well as fiber breaks, after processing glass fiber-reinforced polypropylene. They used either short or long glass fibers. The results showed that, due to the shear in the processing step, long glass fiber-reinforced polypropylene leads to more fiber breakages than short glass fiber-reinforced polypropylene. [16].

In order to comprehensively determine the correlation of process parameters during compounding experimentally, a high level of testing, personnel and material expenditure is required [17,18]. To date, fiber length change in composites has primarily been determined by microscopic observation and X-ray microtomography of the samples [19,20,21,22,23]. To enable significant quality improvements, experimentally validated modeling is required that reaches far beyond the current state of technology and considers the details of the used machines and materials. This would also significantly reduce the experimental effort. The higher the prediction quality of the fiber length distribution of the software used, the more accurate the prediction that can be made about the resulting mechanical behavior for the subsequent application areas.

In this study, short glass fiber reinforced thermoplastics (polypropylene) were produced on two different twin-screw extruders at the University of Paderborn (Kunststofftechnik Paderborn, Paderborn, Germany) and the University of Kassel (Institut für Werkstofftechnik, Kunststofftechnik, Kassel, Germany). First, a comparison of the real and simulated fiber length changes was necessary, based on which, the simulation models and their parameters were adjusted. Subsequently, the results could be adapted to real structures. The influences of the compounding parameters, in particular the different machine and screw configurations, on the real and simulated fiber length change are presented, whereby the real fiber length change was determined by means of dynamic image analysis (Section 2.3). Since the processing was carried out on two different machines, the machine-specific process behavior was of major interest regarding its influence. With the simulation software SIGMA (Simulation of co-rotating twin-screw machines), fiber length, in general, can be calculated. However, the software utilizes idealized machines. As a result, important knowledge could be obtained by comparing the simulation and experimental studies, and the effects of process and material changes on fiber lengths could be determined in advance. This allows a specific adjustment of the machine settings, whereby the components to be manufactured subsequently can be produced without any defects.

## 2. Materials and Methods

### 2.1. Used Materials

For the manufacturing of the test specimens, polypropylene (SABIC^®^ PP 520P, SABIC Polymers, Genk, Belgium) provided by SABIC Polymers (Genk, Belgium) with a glass fiber (e-glass fiber FGCS 3540, Schwarzwälder Textil-Werke, Schenkenzell, Germany) provided by Schwarzwälder Textil-Werke (Schenkenzell, Germany) compounds with fiber content of 20, 30, and 40 wt.% content were manufactured. No additional additives were added to the polymer and fibers in the compounding process. According to the manufacturer, SABIC^®^ PP 520P is a polypropylene (PP) homopolymer grade and can be used for the production of injection-molded or extruded products or can be used as a component of other industrial products. It provides an excellent combination of transparency, stiffness, and high heat resistance. The material has a density of 0.905 to 0.930 g/cm^3^. The melting temperature (Tm) of the material is 160 to 170 °C (peak at 164 °C), and the melt volume rate (MVR) is 10.5 g/10 min (230 °C/2.16 kg) according to ISO 1133.

The short cut glass fibers FGCS 3540 show a diameter of approx. d_f_ = 10 µm and a mean glass fiber length of approx. l_f_ = 3 mm. The glass fiber has a density of 2.53 to 2.55 g/cm^3^. The moisture content in delivery form is a maximum of 0.3%. The fiber sizing content is approx. 1.1%, according to DIN ISO 1887 (sizing content determination by loss on ignition (625 °C)). The moisture content of the materials was checked before processing using a Sartorius Moisture Analyzer MA 100 (MA 100, Satorius, Goettingen, Germany).

### 2.2. Processing

Compounding of the materials was carried out on a Leistritz ZSE 18 (ZSE 18, LEISTRITZ Extrusionstechnik, Nürnberg, Germany) and a Coperion ZSK 25 WLE (ZSK 25 WLE, Coperion, Stuttgart, Germany). Both are co-rotating twin-screw extruders. The Leistritz ZSE 18 was used with two screw configurations. This extruder had a screw diameter of 18 mm, with an L/D ratio of 40. The material was compounded without a nozzle, to avoid further shortening of the fibers due to different nozzle configurations. Therefore, no downstream equipment needed to be considered in the simulation. The screw configuration (SC1) included kneading discs and mixing elements, to mix and distribute the fibers after they had passed through the feeding zone. The screw configuration (SC2) simply consisted of the conveying elements after the fiber feeding zone. The screw speed was set to 200 rpm, and the material throughput was set to 3 kg/h.

A Coperion ZSK 25 was used with one screw configuration. This extruder had a screw diameter of 25 mm with an L/D ratio of 41. The used screw configuration (SC3) simply consisted of the conveying elements after the fiber feeding zone. The screw speed was set to 200 rpm, and the material throughput was set to 9 kg/h. The different screw designs are shown in Figure 1. The kneading discs are shown in blue and the mixing elements in green. The temperatures of the heating zones are listed in Table 1.

### 2.3. Characterization

#### 2.3.1. Dynamic Image Analysis

Dynamic image analysis was used to investigate the influence of fiber length during compounding on various compounders. For the fiber length measurement, the produced pellets were ashed in an oven (600 °C, 6 h). This removed the matrix material polypropylene from the glass fibers for subsequent image analysis. The fiber length distribution was measured using a Sympatec QICPIC/R06 (QICPIC/R06, Sympatec, Clausthal-Zellerfeld, Germany) dynamic image analysis with a wet disperser (MIXCEL unit, Sympatec, Clausthal-Zellerfeld, Germany). The images were acquired at a frame rate of 175 Hz with an M7 magnification. The ISO measurement range varied from 88 µm to 2888 µm. The results represent the change in fiber length after compounding.

#### 2.3.2. Material Characterization for Simulation

The rheological material data analysis required for the simulation, such as the viscosity, thermal conductivity, and p-v-T behavior, were determined using a high-pressure capillary rheometer (RG25, Göttfert, Buchen, Germany). The melting point, enthalpy, and specific heat capacity were determined by differential scanning calorimetry (DSC Q2000, TA Instruments, New Castle, DE 19720, USA). The determined data for the used polymer and the glass fiber were entered into the database PAM (Paderborn Material Database)(PAM 3.0.0, Kunststofftechnik Paderborn, Paderborn, Germany) and exported into the simulation software Sigma (SIGMA 13.0.1, Kunststofftechnik Paderborn, Paderborn, Germany).

#### 2.3.3. Simulation of the Compounding Process

The SIGMA simulation software was used to simulate the compounding process. The simulation software simulates compounding and processing operations on co-rotating twin-screw extruders. In addition to the material data, the barrels and screw configurations of the used extruders (see above) were created in SIGMA. Using models for polymer-filler compounds, the compounding processes were simulated for the matrix–fiber designs and various screw configurations of the different extruders used.

### 2.4. Extrusion Simulation of Fiber Length

The SIGMA simulation software was developed for the design of co-rotating twin-screw extruders. For the further development of SIGMA, Kunststofftechnik Paderborn cooperates with the leading machine and material manufacturers and continuously optimizes the simulation software. For the design of a twin-screw extruder, the user has at his disposal, for example, not only the calculated melting, pressure, and filling degree curves, but also the fiber length degradation of fillers, such as glass fibers or carbon fibers, and the dwell time. These curves can be used to optimize the basic process. For this publication, the calculation of the expected fiber length was of relevance. For this purpose, the real processes with the machines and material data used were entered into SIGMA and subsequently simulated.

## 3. Results and Discussion

### 3.1. Influence of Screw Configuration on the Fiber Length

Figure 2 shows the influence on the fiber length during the production of polypropylene + 20 wt.% glass fibers (PP GF 20) with different screw configurations (SC1, SC2, SC3) in the extrusion process. The distribution density is given as a relative frequency. The results show that the production of the materials with the SC1 and SC3 screw configurations led to a significant shortening of the glass fibers. Considering the relative fiber distribution, these two screw configurations showed the shortest fibers after the extrusion process, with a length of 10–30 µm. Analogous to this, the results of SC2 showed similar tendencies. In the range from 400 to 600 µm, a further local maximum in the fiber length was detectable for the SC1 screw configuration. The local maximum for screw configuration SC2 was around 800 µm, and there was no further local maximum for screw configuration SC3. Nevertheless, the total fiber shortening was slightly lower in comparison with the other two screw configurations.

In a comparison of the results shown in Figure 3 (PP + GF 20), the results of polypropylene with 30 wt.% glass fibers also showed that the highest shortening of the fibers occurred in the production with the screw configurations SC1 and SC3. In addition, it is also shown here that there were no significant differences between these two screw configurations. Moreover, it can be seen here that the fiber shortening with the SC2 screw configuration was slightly lower, and significantly fewer short fibers resulted after processing compared to the SC1 and SC3 screws. Overall, the behavior and fiber distribution were very similar to the values of PP + GF 20 (cf. Figure 2).

Figure 4 shows the influence on the fiber length during the production of polypropylene + 40 wt.% glass fibers (PP GF 40) with different screw configurations (SC1, SC2, SC3). The larger amount of glass fibers resulted in a further shortening of the fibers, so that overall shorter fibers were measured for the screw configurations SC1 and SC3. Again, significantly less fiber shortening resulted when using screw SC2 compared to both screws SC1 and SC3.

### 3.2. Simulated Fiber Length

The simulated fiber length is shown graphically in Figure 5. The simulated fiber length decreased with increasing glass fiber content. With a glass fiber content of 20 wt.%, the minimum was 516 µm for screw configuration SC1 and the maximum was 551 µm for screw configuration SC3. If the glass fiber content was increased to 30 wt.%, a maximum fiber length of 490 µm was achieved for screw configuration SC3. The minimum fiber length was 470 µm with screw configuration SC1. With a glass fiber content of 40 wt.%, only a maximum fiber length of 439 µm was achieved with screw configuration SC3. The minimum fiber length was about 426 µm and was calculated for both screw configurations SC1 and SC2. On average, the fiber lengths of screw configuration SC3 were 3.73% longer than the fiber lengths of screw configuration SC2. With screw configuration SC1, the resulting fiber lengths were on average 0.71% shorter than the fiber lengths of screw configuration SC2. The differences between the simulated fiber lengths decreased with increasing fiber content. At a fiber content of 40 wt.%, the simulated fiber lengths of screw configurations SC1 and SC2 no longer differed.

In comparison to Figure 5, Figure 6 shows the simulated fiber length at a 20 wt.% fiber content along the axial extruder position. The fiber length decreased rapidly shortly after being added to the extruder. Already, after approximately 15 mm, the fiber length had decreased from 3 mm to 2 mm. After approximately 82.5 mm, the fiber length was already 1 mm for screw configurations SC1 and SC2. With screw configuration SC3, the fiber length dropped below 1 mm only after approx. 107.5 mm. The fiber length decreased slightly at the end of the compounding process.

The effect of varying the fiber content with otherwise constant process parameters on the curve of the simulated fiber length is shown in Figure 7. As in Figure 6, the curve is very steep at the beginning and becomes flatter towards the end of the screw. The fiber content influences the slope of the fiber length curve. With a larger fiber content, the fiber length curve is steeper at the beginning and is below the values of the curves with a smaller fiber content.

Figure 8 shows the different fiber length distributions for the screw configuration SC1 with different fiber contents. The local maximum in fiber length distribution changed from approx. 550 µm to approx. 450 µm with a higher fiber content.

The results suggest that the measured particle sizes do not correspond to reality. The large proportion of particle sizes shorter than 50 µm may contain not only glass fibers but also remnants of ashing particles and dust particles. If the particle sizes in the iso-measurement range between 100 and 2888 µm are examined more closely, it is noticeable that both the SC2 and SC3 screw configurations have a significantly smaller proportion of fiber lengths over the entire range than the SC1 screw configuration. This effect is independent of the fiber content. Nevertheless, the resulting local maxima of screw configuration SC1 are ahead of the local maxima of screw configurations SC2 and SC3. These results correspond to the simulation results of SIGMA. The simulation results do not exactly match the experimental results for smaller fiber fractions. However, as the fiber content increases, the difference between the simulation result and the experimental result decreases. The difference between the screw configurations SC2 and SC3 is not as clear in the measurement as it is in the simulation. The resulting fiber length also decreased with increasing fiber content. This effect was more pronounced for the three screw configurations when the glass fiber content was increased from 30 to 40 wt.% then it was with the increase from 20 to 30 wt.%.

The resulting fiber lengths and distributions in this study were slightly above the fiber lengths and distributions documented in the literature. Ghanbari et al. [24] investigated the fiber lengths and distributions at 75 and 400 1/min, using a similar screw design. The measured fiber lengths ranged from about 260 µm at 75 1/min to about 50 µm at 400 1/min. Whereas, in Bumm et al. [25], the resulting fiber lengths were much closer to the values from this study. Nevertheless, the fiber lengths were slightly above the expected range specified in the literature and those calculated in the SIGMA simulation software. The decrease in glass fiber length due to the kneading blocks after addition of the glass fibers can be clearly seen in the resulting fiber lengths. As mentioned in [25], fiber length decreases during compounding with kneading blocks. The SC1 screw configuration consistently produced lower resulting fiber lengths than the other two screw configurations. Furthermore, the resulting fiber length decreased with increasing glass fiber content. This behavior was previously investigated by Durmaz and Aytac [15] and was consistent with the experimental results in this study. The simulated curve of the fiber lengths along the axial extruder position is similar to the curves presented in the literature. The fiber lengths decreased rapidly directly after the addition of the fibers. In the further progression, the fiber length continued to decrease, but no longer as rapidly as at the beginning [13,25].

Thus, the resulting fiber lengths differed for the same process and boundary conditions, not only due to a different screw configuration, but also for identical screw configurations on different twin-screw extruders. These results were evident in both the simulation and experimental results.

## 4. Conclusions

In this article, short glass fiber reinforced thermoplastics (polypropylene) were produced on two different twin-screw extruders, with three different screw configurations (SC1, SC2, SC3) at the University of Paderborn (Kunststofftechnik Paderborn, Paderborn, Germany) and the University of Kassel (Institut für Werkstofftechnik, Kunststofftechnik, Kassel, Germany). Fiber shortening, which can result during processing, was simulated using the SIGMA simulation software. In addition, real experiments were carried out, in order to determine the fiber length changes due to the compounding process.

The different particle size distributions showed that the resulting particle size distribution depended, not only on the screw configuration, but also on the machine size. The screw configuration with kneading blocks (SC1) led to the smallest fiber lengths. Despite an identical screw configuration of SC2 and SC3, the results of the fiber lengths of both screw configurations were different from one to the other. Both in simulation and in practice, the resulting fiber lengths differed. Nevertheless, the resulting fiber lengths of SC2 and SC3 were both greater than those of screw configuration SC1. With the different particle distributions of the SC2 and SC3 screw configurations, the question arises as to whether the different course of the particle distribution is also dependent on other machine parameters, in addition to the change in machine size. For example, manufacturer-specific gap dimensions between the barrel wall and the screw could have a recognizable influence on the resulting particle size. In order to be able to exclude these machine-related influences or, if necessary, to specify them in more detail, further investigations on several twin-screw extruders with different manufacturer-specific gap dimensions are necessary.

## Figures and Tables

**Figure 1 polymers-14-03113-f001:**
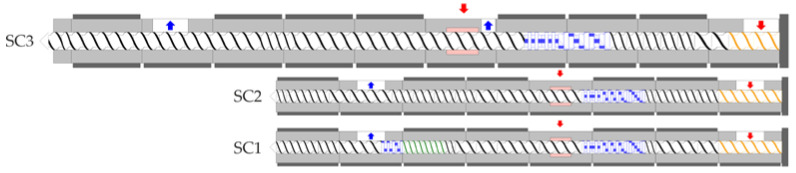
Used screw configurations in the different twin-screw extruders.

**Figure 2 polymers-14-03113-f002:**
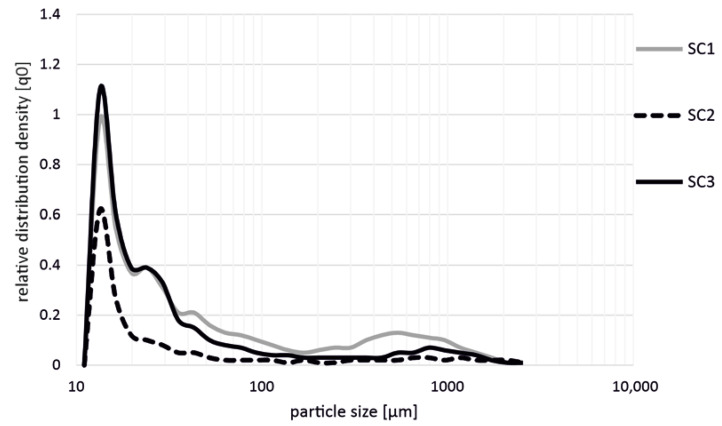
Influence on fiber length during the production of PP GF 20 with different screw configurations (SC1, SC2, SC3) in the extrusion process.

**Figure 3 polymers-14-03113-f003:**
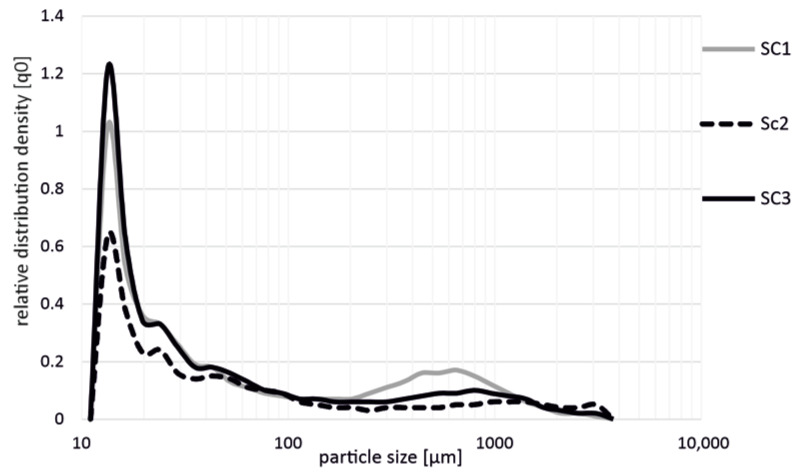
Influence on fiber length during the production of PP GF 30 with different screw configurations (SC1, SC2, SC3) in the extrusion process.

**Figure 4 polymers-14-03113-f004:**
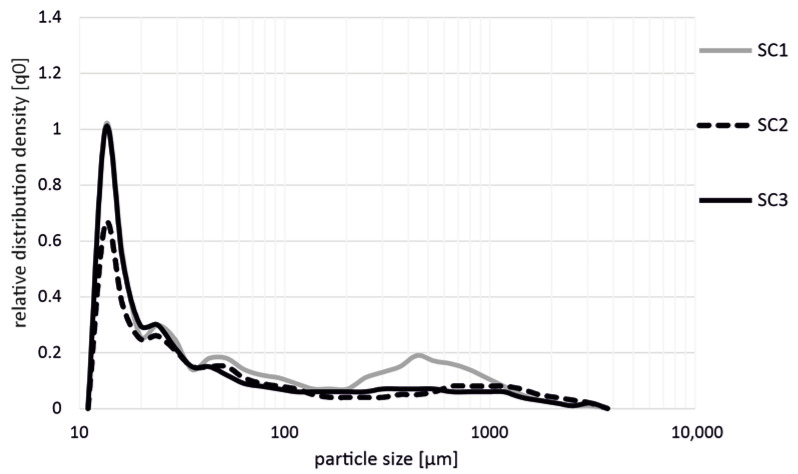
Influence on fiber length during the production of PP GF 40 with different screw configurations (SC1, SC2, SC3) in the extrusion process.

**Figure 5 polymers-14-03113-f005:**
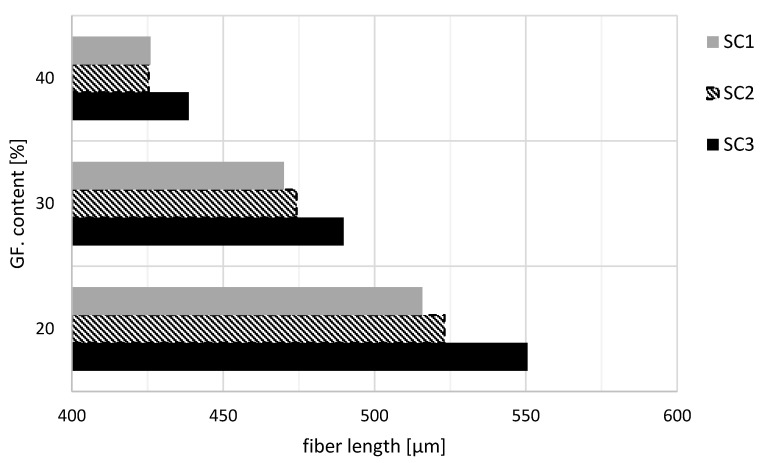
Results of the simulation of the fiber lengths for the glass fiber contents of 20, 30, and 40 wt.% for the three different screw configurations.

**Figure 6 polymers-14-03113-f006:**
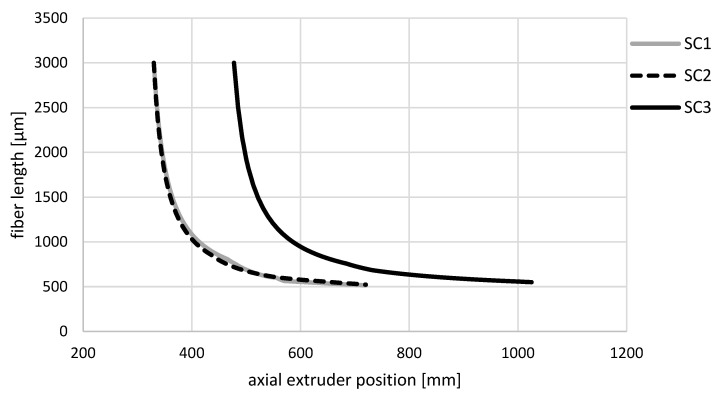
Simulated fiber length curves for the glass fiber content of 20 wt.% for the three different screw configurations along the axial extruder position.

**Figure 7 polymers-14-03113-f007:**
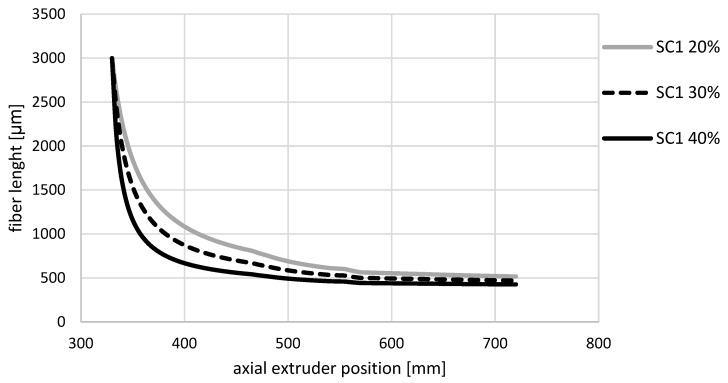
Simulated fiber length curves for the glass fiber content 20, 30, and 40 wt.% for the screw configuration SC1 along the axial extruder position.

**Figure 8 polymers-14-03113-f008:**
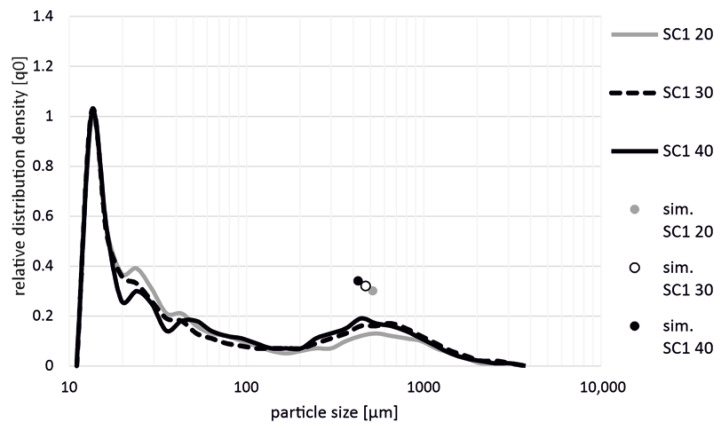
Illustration of the fiber length distributions as a function of fiber content for screw configuration SC1.

**Table 1 polymers-14-03113-t001:** Temperatures used in the compounding process in the different twin-screw extruders.

Process Temperatures (°C)	Leistritz ZSE 18	Coperion ZSK 25
Feeding Section	50	25
Zone 1	160	160
Zone 2	180	180
Zone 3	180	180
Zone 4	180	180
Zone 5	180	180
Zone 6	180	180
Zone 7	180	180
Zone 8		180
Zone 9		180

## Data Availability

Not applicable.

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
