# Peer review of "Influence of Processing Glass-Fiber Filled Plastics on Different Twin-Screw Extruders and Varying Screw Designs on Fiber Length and Particle Distribution"

_polymers, 2022, doi:10.3390/polym14153113_

Round 1

Reviewer 1 Report

This paper is very technical, it should be published in a processes or manufacture journal. Even so, it should be improved. You can find below some comments to improve the paper quality.
1.-Pictures of different model of screws need to be inclued.
2.-There are no definitive conclusions, some parameters, as the authors themselves comment, were not taken into account.
3.-A factorial desing of experimentw will be able to give information, not only of the parameter effects, but the interaction between them.

Author Response

1.-Pictures of different model of screws need to be inclued.

There is already a graphic with the different screw designs in the article (compare figure 1). Nevertheless, additional information has been added to describe the individual screw elements shown.

2.-There are no definitive conclusions, some parameters, as the authors themselves comment, were not taken into account.

Work was done on the conclusion to highlight the results more clearly.

3.-A factorial desing of experimentw will be able to give information, not only of the parameter effects, but the interaction between them.

A factorial experimental design cannot be implemented in a short time. Changing the manufacturer-specific dimensions implies that enough equipment from different manufacturers is available. This is planned in further investigations.

Reviewer 2 Report

Comments:

The paper is interesting for Polymers journal being focused on polymer processing topic.

1. Show the novelty and added value of the paper with respect to existent literature on the topic.

2. Abstract: I would replace positive properties with something else like good or valuable/interesting etc.

3. Introduction: I would replace positive properties like in the abstract.

4. The number of references is quite low for a scientific paper. You could add more references in the introduction or Results and discussion part.

5. The paper lacks a discussion section where the resuls should be correlated with similar data in literature.

6. Did you add some other additives for processing or only PP and glass fibers? Does the commercial PP contain processing additives like stabilizers for example?

7. "The moisture content in delivery form is maximum 0.3%." How did you check this aspect?

Author Response

1. Show the novelty and added value of the paper with respect to existent literature on the topic.

The results are now compared with results from the literature

2. Abstract: I would replace positive properties with something else like good or valuable/interesting etc.

Has been adjusted.

3. Introduction: I would replace positive properties like in the abstract.

Has been adjusted.

4. The number of references is quite low for a scientific paper. You could add more references in the introduction or Results and discussion part.

Additional literature has been included for the discussion of the results.

5. The paper lacks a discussion section where the resuls should be correlated with similar data in literature.

A discussion section has been added comparing the results with other published results.

6. Did you add some other additives for processing or only PP and glass fibers? Does the commercial PP contain processing additives like stabilizers for example?

The information was added. No additives were added.

7. "The moisture content in delivery form is maximum 0.3%." How did you check this aspect?

The information was added including the information of the machine used for this analysis.

Reviewer 3 Report

The paper aims to Influence of processing glass-fiber filled plastics on different twin-screw extruders and varying screw designs on fiber length and particle distribution. Fiber-reinforced thermoplastics are mainly manufactured by injection molding and extrusion. To enable significant quality enhancement, experimentally validated modeling is required. Short glass fiber reinforced thermoplastics are produced on two different twin-screw extruders. The machine-specific process behavior is of major interest regarding its influence. In this way, important knowledge can be obtained by comparing simulation and experimental tests, and the effects of process and material changes on the fiber lengths can be determined in advance.  In general, the paper can be recommended for publication with revisions. Some comments for revisions are given below:

1.     The language of manuscript should be further polished.

2.      Although the work of this manuscript has valuable contribution to the science and practical application, the author should emphasize the significance of the work in the manuscript.

3.      There are some grammatical errors in the manuscript that need to be modified.

4.      Refine the article and Refined abstract, concisely and accurately elaborate the content of the manuscript, reduce redundant expressions.

5.      Some of the latest references are recommended reference in Introduction. For example:

Yun Chen, Bin Xie, Junyu Long, Yicheng Kuang, Xin Chen, Maoxiang Hou, Jian Gao, Shuang Zhou, Bi Fan, Yunbo He, Yuan-Ting Zhang, Ching-Ping Wong, Zuankai Wang, Ni Zhao. Interfacial laser induced graphene enabling high performance liquid-solid triboelectric nanogenerator, Advanced Materials, 2021, 33, 2104290.

Z. Liu, J. Li, X. Liu. (2020) Novel functionalized BN nanosheets/epoxy composites with advanced thermal conductivity and mechanical properties, ACS Applied Materials & Interfaces, 12(5), 6503-6515

Author Response

  1. The language of manuscript should be further polished.
    The written work was checked again for linguistic and grammatical errors.
  2. Although the work of this manuscript has valuable contribution to the science and practical application, the author should emphasize the significance of the work in the manuscript.
    The Discussion and Conclusion have been revised to better highlight significance.

  3. There are some grammatical errors in the manuscript that need to be modified.
    See reply to 1.

  4. Refine the article and Refined abstract, concisely and accurately elaborate the content of the manuscript, reduce redundant expressions.
    The summary was revised and shortened. The results of the investigations have also been incorporated.

  5. Some of the latest references are recommended reference in Introduction.
    The references mentioned were analyzed. Thematically, the references mentioned are too far away from glass fiber reinforcement compounding. Accordingly, the references could not be meaningfully included in the introduction or the article.

Round 2

Reviewer 1 Report

Although unquestionably, the paper has improved after the corrections, my opinion is the same. This paper is still a technical paper, there is no scientific study that shows or rules out changes in the physical-chemical properties of the composite material or the fibers. For example the crystallinity of polypropylene or  its melting point, are variables normally studied after extrusion process. There are also no images from dynamic image analysis, showing the change in fiber length after preparation. Not even a scientific discussion of why changes are taking place.

Author Response

This article is not intended to deal with the physical-chemical properties of the composite material. For this reason, no investigations were carried out regarding the change in properties (mechanical or chemical) of the PP after the extrusion process. The fact that fiber length degradation takes place and thus the mechanical properties can be undesirably changed has already been investigated in various studies and is therefore only a minor part of this article. Instead, this article focuses on fiber length degradation and fiber length distribution on different machines under the same general conditions.
It was deliberately decided not to present the individual images of the image analysis, since no or only very little added value was seen in these individual images for this article.